# Consistent Zero-Shot Imitation with Contrastive Goal Inference

Kathryn Wantlin [1]   Chongyi Zheng [1]   Benjamin Eysenbach [1]

## Abstract

Zero-shot imitation learning requires an agent to reproduce expert behavior from a single demonstration without additional environment interaction or gradient updates at test time. We introduce Contrastive Inverse Reinforcement Learning (CIRL), a self-supervised framework for pretraining zero-shot imitation agents. Our methods rests on a key observation that many useful tasks can be summarized by a single goal state. We can thus convert the multi-task inverse RL problem into a more tractable goal-inference problem, and utilize state-of-the-art goal-conditioned RL methods to recover a policy that reaches the goal. During pre-training, CIRL jointly employs three components to learn without any rewards or demonstrations: (1) a variant of contrastive RL designed to learn maximum-entropy goal-conditioned policies, (2) an automatic goal proposal mechanism (GoalKDE) that drives exploration, and (3) a mean-field variational model that performs amortized goal inference from trajectories. We prove that this procedure consistently recovers the demonstrator's intent by accounting for the relative difficulty of reaching different states and show how structurally similar prior work may otherwise fail to infer the correct reward. Experiments on goal-conditioned and standard reward-maximizing control tasks show that CIRL outperforms prior zero-shot imitation methods, supporting the expressiveness of goals as a compact summary of behavior.

## 1. Introduction

Today's AI agents, whether in language or robotics, are trained primarily by mimicking human demonstrations. But, in the same way that children conduct a large degree of learning in an unsupervised (adult-free) fashion (Gweon & Schulz, 2019; Gopnik, 2020; Stahl & Feigenson, 2015; Poli et al., 2025; Bonawitz et al., 2011), how might AI agents develop a foundation of knowledge through exploration and play, rather than through mimicry? In this paper, we study the setting where agent pretraining is done with no demonstrations, no internet-scale data, and no rewards, but rather through self-supervised interaction (Ma et al., 2022; Wu et al., 2018; Eysenbach et al., 2018; Pathak et al., 2017; Mendonca et al., 2021). The agent proposes goals, attempts to reach them, and learns from these self-collected data. After training, this agent is assessed by its ability to imitate: given a demonstration, the agent uses a (learned) inverse RL module to infer the demonstrator's goal, and then uses the (learned) goal-conditioned policies to reach that goal. Our problem setting is thus *zero-shot imitation learning (IL)*, where we would like to infer behaviors from a single demonstration without additional gradient updates (Pirotta et al.; Pathak et al., 2018; Jang et al., 2021).

It is unclear whether today's recipe for building generative AI foundation models will be directly applicable to *interactive* settings. Though the premise of agents is online exploration or action, generative models are primarily built by optimizing self-supervised objectives on input data (Bommasani et al., 2021) collected offline by humans. In robotics, policies are typically constructed by either mimicking human demonstrations (Chi et al., 2023; 2024; Octo Model Team et al., 2024; Reed et al., 2022) or maximizing human-specified rewards (Silver et al., 2016; Wurman et al., 2022). These approaches do have an explicit notion of action, but agents typically practice on a limited set of tasks and are not required to infer a demonstrator's intention. The key focus in our paper is on purely self-supervised pretraining for agentic systems with interactive exploration and inverse RL that accounts for the relative difficulty of different tasks (Eysenbach et al., 2020; Ziebart et al., 2008; Ng & Russell, 2000).

Related work in inferring intentions projects a demonstration onto a hypothesis space of reward functions and then trains a general-purpose zero-shot RL policy over this space of rewards (Touati & Ollivier, 2021; Wu et al., 2018). We make the additional key observation that many tasks can be described in terms of goals, such as navigation or manipulation tasks (Brockman et al., 2016). In these settings,

---

[1]Department of Computer Science, Princeton University, Princeton, NJ, USA. Correspondence to: Kathryn Wantlin <kw2960@princeton.edu>.

*Proceedings of the $43^{rd}$ International Conference on Machine Learning*, Seoul, South Korea. PMLR 306, 2026.

goals are described by the agent's state, and we can imagine extending the state space to capture more complex reward functions. Tasks where the necessary actions are more complex or hierarchical, such as cooking a recipe in a kitchen, could also be described by a high-dimensional observational state, natural language, or multiple sub-goals. Maintaining a prior that tasks can be described via goals allows us to apply state-of-art goal conditioned reinforcement learning (GCRL) methods to a reduced hypothesis space of reward functional forms (Kaelbling, 1993; Schaul et al., 2015; Andrychowicz et al., 2017). While Zheng et al. (2026) has shown how to convert any arbitrary reward-maximization MDP problem into a goal-conditioned MDP while preserving policy optimality, this conversion requires an expansion of the state space that could be nontrivial to design in practice. Previous imitation learning and inverse RL methods are restricted by their hypothesis space of reward functions, and our method is limited by the state space we define over the environment. However, simply learning to reach goals has been shown to drive strong exploration and skill acquisition, even with very sparse rewards and high dimensional environments (Liu et al., 2024).

This leads us to the key question: how effectively can we imitate when we use goals as a compact summary of behavior? We show experimentally that by projecting behavior onto the restricted space of goal-conditioned reward functions, we can more efficiently summarize and imitate a range of tasks from important benchmarks. Therefore, we re-imagine solving the zero-shot imitation learning task by first inferring the expert's goal and then commanding a zero-shot goal-conditioned RL policy to this inferred goal. We start by assessing our method on goal-reaching tasks, and then evaluate on reward-maximization tasks not tied to particular goal states. Our main contributions is as follows:

- We propose a contrastive inverse reinforcement learning algorithm (CIRL) for self-supervised pretraining of interactive agents that extends contrastive reinforcement learning (CRL) methods to the MaxEnt RL setting and includes automatic goal sampling during pretraining. Training involves exploration and learning via trial and error, yet requires no demonstrations, no rewards, and no preferences.

- Unlike some structurally similar methods, we prove that our method is consistent: it correctly infers the user's goal using inverse RL, accounting for the relative difficulty of reaching different goals.

- Empirically, we show that our method performs effective autonomous exploration and rapid adaptation in the standard URLB benchmark (Laskin et al., 2021), outperforming prior zero-shot imitation and zero-shot RL methods.

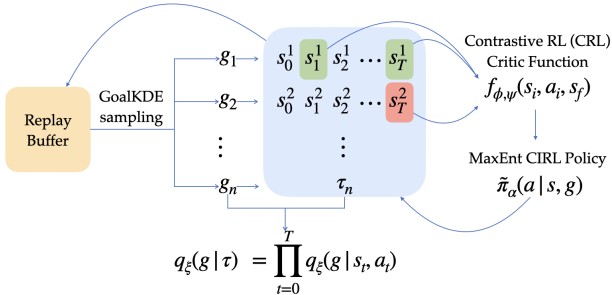

$$q_\xi(g\,|\,\tau) \;=\; \prod_{t=0}^{T} q_\xi(g\,|\,s_t, a_t)$$

*Figure 1.* **CIRL algorithm components:** (1) An automatic goal-proposal mechanism, GoalKDE, that drives exploration by sampling under-visited states from the replay buffer. (2) A variational model that performs amortized goal inference from a trajectory using a mean field form. (3) A maximum-entropy goal-conditioned policy trained with contrastive RL. At test time, the agent will see a trajectory $\tau$, infer $\hat{g} \sim q_\xi(g|\tau)$ and execute the imitation policy $\tilde{\pi}_\alpha(\cdot\,|s, \hat{g})$.

## 2. Related Work

We turn to GCRL benchmarks to test our hypotheses for goal-conditioned zero-shot IL. Several state-of-art methods on goal-reaching RL use variants of temporally contrastive objectives to learn representations and policies, and extend successor feature-based methods to high dimensional environments (Wang et al., 2023; Eysenbach et al., 2022; Myers et al., 2024). However, prior methods are limited in their assumption of access to the test-time distribution of goals, focus on the offline setting, or a hand-designed exploration policy (Pathak et al., 2018; Eysenbach et al., 2022). Given the strength of these methods in RL settings, we naturally ask whether their representations would be useful for imitation, and whether we can extend them to also learn to command their own goals. We build off the JaxGCRL benchmark to test our ideas with the Contrastive Reinforcement Learning (CRL) algorithm on a well-designed suite of tasks (Bortkiewicz et al., 2025).

Our work builds on a rich literature on URL, which uses reward-free data to improve performance and generalization of RL algorithms. Some methods focus on extracting reusable representations (Wu et al., 2018; Ghosh et al., 2023; Ma et al., 2022; Blier et al., 2021; Sikchi et al., 2024). However, some of these methods that offer strategies for inferring rewards from demonstrations contain a faulty assumption that matching the expert distribution is sufficient for inferring the demonstrator's policy without accounting for the partition function over tasks (Touati & Ollivier, 2021; Sikchi et al., 2024). Other works focus on unsupervised discovery of diverse skills (Gregor et al., 2016; Machado et al., 2017; Eysenbach et al., 2018; Sharma et al., 2019; Eysenbach et al., 2021; Klissarov & Machado, 2023; Zahavy et al., 2022; Park et al., 2023a;b; Zheng et al., 2024;

Wang et al., 2024). While some of these exhibit zero-shot policy inference capabilities from rewards or goals, they are not designed to perform zero-shot imitation from demonstrations and do not necessarily discover all possible skills (Zheng et al., 2024; Eysenbach et al., 2021). Methods of online unsupervised exploration for pretraining policies tackle problems under similar assumptions as our work, but do not handle zero-shot inference given trajectories (Pathak et al., 2017; 2019; Mendonca et al., 2021; Rajeswar et al., 2022). There has also been significant development of offline unsupervised pre-training methods, but these could suffer under poor exploration and do not focus on the interaction between exploration and policy learning in unfamiliar environments. Our method is highly connected to prior work on learning universal, high dimensional successor representations, with applications to both online and offline settings (Dayan, 1993; Barreto et al., 2016; Borsa et al., 2018; Ma et al., 2018; Touati & Ollivier, 2021; Touati et al., 2022; Pirotta et al.). Like these works, our method contains an inductive bias influencing which tasks we focus on, namely those that are goal-reaching. We show that this restriction becomes particularly useful for inverse RL, even for arbitrary reward functions.

Approaches to zero-shot imitation learning combine approaches to inverse RL and exploration/data collection to solve the problem. We'll discuss these individual components first and then discuss key prior methods for zero-shot imitation.

**Inverse RL** Achieving general, adaptable agents is challenging via reward engineering and may lead to unintended behaviors (Amodei et al., 2016). Thus, we turn to learning from demonstrations (LfD), assuming we have access to limited data from an expert (Finn et al., 2016; Fu et al., 2018; Pirotta et al.; Yu et al., 2019). The main approaches to LfD are behavioral cloning (BC) and inverse reinforcement learning (IRL). BC casts learning an imitation policy as a supervised learning problem. While BC can work well in practice, it suffers from poor performance under distributional shift and can overfit its expert demonstrations (Ross et al., 2011; Pomerleau, 1988; Bojarski et al., 2016). IRL attempts to infer reward functions/corresponding policies from demonstrations (Ng & Russell, 2000). Since the reward inference problem is inherently under-specified, a common modeling choice is the Maximum Entropy assumption, which assumes that expert demonstrations select actions to maximize both the sum of expected discounted rewards and the entropy of the distribution of actions over states (Ziebart et al., 2008). Extensions such as GAIL, AIRL, and GCL were developed to use deep function approximators for single-task IRL (Ho & Ermon, 2016; Fu et al., 2018; Finn et al., 2016). Current multi-task/meta IL algorithms can be categorized as gradient-based or context-based (Chen

et al., 2023). Gradient-based approaches, such as (Finn et al., 2017; Yu et al., 2018) combine meta-learning with IL to recover a policy, but at inference time, require a one-shot gradient step to adapt to a new task whereas our method adapts zero-shot. Context-based approaches such as SMILE and PEMIRL learn a latent variable to represent the task contexts and train a context-conditioned policy that can be applied zero-shot to new tasks (Seyed Ghasemipour et al., 2019; Yu et al., 2019). Our approach is similar (encoding goals as a form of context) but takes this one step further by proving that the multi-task IRL problem can actually be reduced to a purely goal-inference problem when we our expert optimizes a goal-conditioned reward function. Therefore, we can use zero-shot RL algorithms to recover policies without loss of performance instead of using less stable adversarial methods.

**Exploration** While BC and IRL can be performed on offline datasets, we would prefer to enable zero-shot imitation through purely online methods that can be applied out-of-the-box in novel environments. This requires our IL agent to perform its own exploration, which CRL currently does not support (Eysenbach et al., 2022). For our goal-conditioned setting, automatic goal sampling enables us to autonomously generate training objectives. Goal sampling approaches broadly fall into two categories: adversarial methods and distribution-based methods. Adversarial methods such as ASP and GoalGAN introduce a second policy for sampling goals (OpenAI et al., 2021; Florensa et al., 2018). While effective for simple domains, these methods can struggle with high-dimensional goal spaces and require careful balancing of the adversarial training process. State distribution approximation methods such as Skew-Fit, EDL, VUVC, RIG, MEGA, and DISCERN control the probability of selecting a goal via the empirical state visitation density, usually trying to cover the full state space with exploration (Pong et al., 2020; Campos et al., 2020; Kim et al., 2023; Nair et al., 2018; Pitis et al., 2020; Warde-Farley et al., 2018). Our method, GoalKDE, adopts a simple form of MEGA, although more complex methods could also be benchmarked in future work.

**Zero-Shot Imitation Learning** BC-Zero addresses multi-task zero-shot imitation by scaling diverse, human-in-the-loop data collection and training a single task-conditioned behavior-cloned policy that can execute novel text instructions at test time (Jang et al., 2021). However, unlike our method, BC-Zero gathers task-labelled expert data via teleoperation and requires human interventions in a DAgger-style loop, whereas our method trains purely online and collects its own data using a self-supervised objective and exploration. Zero-Shot Visual Imitation uses goal-conditioned policies to imitate experts trained via a model-based forward consistency loss (Pathak et al., 2018). However, un-

like our work, they hand-devised an exploration policy to generate data for model-based training, whereas our data collection is fully self-supervised for model-free training. Forward-Backward (FB) Representations and RLZero enable zero-shot imitation through matching the demonstrator's state visitation distributions (Touati & Ollivier, 2021; Pirotta et al.; Sikchi et al., 2024). However, we prove for the FB representation that without accounting for the partition function, this method leads to systematic misidentification of the demonstrator's true policy.

## 3. Preliminaries

**Definition 3.1.** The **zero-shot imitation learning** problem assumes we are given a single expert trajectory $\tau = (s_0, a_0, ..., s_T, a_T)$ at inference time, generated by some unknown expert policy $\pi_E$ with trajectory distribution $p_{\pi_E}(\tau)$. No reward function is available. We must produce a policy $\hat{\pi}_{CIRL} \in \Pi$ that successfully reproduces the behavior of $\pi_E$ defined by its unknown reward function, thereby achieving low regret. $\hat{\pi}_{CIRL}$ should be inferred from $\pi_E$ with no additional environment interaction, test-time data, or gradient updates.

To solve this problem, we will model the environment as a goal-conditioned MDP, defining a reward function that depends on a goal and thereby assuming that expert policies $\pi_E$ have behaviors that can be described as goal-reaching. Then, we can infer the reward function associated with $\pi_E$ via MaxEnt IRL. To do this, we will infer the goal $\hat{g}$ associated with $\pi_E$, and command a goal-conditioned policy to $\hat{g}$ that is trained with CRL. In the subsequent sections, we will prove that performing MaxEnt IRL with a goal-conditioned reward is equivalent to performing goal inference. We operate in the pure online RL setting, assuming no access to offline expert data or the test-time goal distribution during pretraining, a departure from CRL's oracle assumptions.

### 3.1. Contrastive RL

We define a goal-conditioned MDP by a tuple $(S, A, G, P, r, \rho)$, where $S$ is the state space, $A$ is the action space, $G$ is the goal space (equivalent to the state space in our formulation); $p : S \times A \times S \to [0, 1]$ describes the transition probabilities between states; $r : S \times A \times G \to \mathbb{R}$ is a goal-conditioned reward function, defined as $r(s_t, a_t, g) = (1 - \gamma)p(s_{t+1} = s_g \mid s_t, a_t) = r_g(s_t, a_t)$, for some discount factor $\gamma$; $\rho_0(s_0)$ specifies the initial state distribution, and $p(g)$ specifies some test-time distribution over goals. We use $\tau$ to define a finite horizon trajectory as a sequence of states and actions: $\tau = (s_0, a_0, \cdots, s_T, a_T, )$, and write the likelihood of a trajectory under policy $\pi$ as $p(\tau) = \rho_0(s_0) \prod_t p(s_{t+1} \mid s_t, a_t) \pi(a_t \mid s_t)$. We also define the discounted future state $s_f$ occupancy measure (density) of goal-conditioned policy $\pi : \mathcal{S} \times \mathcal{G} \to \Delta(\mathcal{A})$ as

$p_\gamma^\pi(s_f|s, a, g) = (1 - \gamma) \sum_{t=0}^\infty \gamma^t p_t^\pi(s_t \mid s, a, g)$ and the marginal distribution as $p_\gamma^\beta(s_f) = \int p^\beta(s, a) p_G(g) p_\gamma^\pi(s_f \mid s, a, g) ds da dg$, where $\beta : \mathcal{S} \to \mathcal{A}$ is the behavioral policy. Using the contrastive RL algorithm, we can estimate the discounted state occupancy using Noise Contrastive Estimation (Oord et al., 2018) and obtain the critic function $f_{\phi,\psi}^\star(s, a, g) = \|\phi(s, a) - \psi(g)\|_2 = \log \frac{p_\gamma^\pi(s_f|s,a,g)}{p_\gamma^\beta(s_f)} = \frac{1}{p_\gamma^\beta(s_f)} \cdot Q_{s_f}^{\pi(\cdot|\cdot)}(s, a)$, where $Q_{s_f}^\pi(s, a) \triangleq \mathbb{E}_{\pi(\tau|s_f)} \left[ \sum_{t'=t}^\infty \gamma^{t'-t} r_{s_f}(s_{t'}, a_{t'}) \mid s_t = s, a_t = a \right]$ (Eysenbach et al., 2022).

### 3.2. Maximum Entropy Inverse Reinforcement Learning (MaxEnt IRL)

We will use the MaxEnt IRL framework to infer reward functions and policies from expert demonstrations. This framework assumes that demonstrations come from a MaxEnt RL policy $\tilde{\pi}^* = \arg\max_\pi \mathbb{E}_{\tau\sim\pi} \left[ \sum_{t=0}^T (r_g(s_t, a_t) + \alpha\mathcal{H}(\pi(\cdot \mid s_t))) \right]$ where $\alpha$ is an optional parameter to control the trade-off between reward maximization and entropy maximization. Without loss of generality, we can assume $\alpha = 1$ for notational simplicity. The trajectory likelihood under the optimal maximum entropy policy is then $p^\star(\tau = \{s_{0:T}, a_{0:T}\} \mid g) = \frac{1}{Z_g} \left[ \rho_0(s_0) \prod_{t=0}^T p(s_{t+1} \mid s_t, a_t) \right] \exp\left( \sum_{t=0}^T r_g(s_t, a_t) \right)$, where $Z_g = \int \rho(s_0) \prod_t P(s_{t+1} \mid s_t, a_t) e^{r_g(s_t, a_t)} d\tau$. We can then define the MaxEnt IRL problem as $\min_{g'} \mathbb{E}_{p(g)} [D_{\mathrm{KL}}(p_E(\tau|g) \| p^\star(\tau = \{s_{0:T}, a_{0:T}\} \mid g'))]$.

### 3.3. Goal Inference

The MaxEnt IRL problem involves inferring reward parameters from a demonstration, and our reward functions are completely parameterized by goals $g$. Therefore, we will perform inference to recover the latent goal of an actor from observed data. Applying Bayes' Rule to the trajectory likelihood of a MaxEnt RL policy, the posterior distribution over goals is $p^\star(g \mid \tau) = \frac{p^\star(\tau|g)p(g)}{p(\tau)} \propto p(g)e^{\sum_t r_g(s_t, a_t) - \log Z_g}$. The partition function $Z_g$ is important for inferring goals, since it gives us a notion of average reward collected along all possible trajectories for a given reward function $r_g(s, a)$. If an expert demonstration collects more reward than this average over trajectories, it is more likely that the demonstration is associated with this particular goal (Eysenbach et al., 2020). The partition function is difficult to estimate, so we will instead fit a variational posterior $q_\xi(g|\tau)$ to perform goal inference (Dragan et al., 2013; Zurek et al., 2021).

# 4. Method

Our algorithm, CIRL, consists of the following components: (1) self-supervised contrastive RL pretraining to learn maximum entropy soft Q-values and a corresponding goal conditioned policy, (2) a goal inference model to learn the variational posterior, and (3) automatic goal sampling during pretraining. Our key contribution is in using goal inference and a goal-conditioned reward to couple IRL with CRL for a successful online imitation learning algorithm. However, certain components, such as the specific goal sampling method, could be substituted.

## 4.1. Maximum Entropy Contrastive Reinforcement Learning

We build an extension of contrastive reinforcement learning under the Maximum Entropy assumption. While CRL just learns the sum of discounted future rewards, we also need to estimate the sum of discounted future entropy to optimize the MaxEnt RL objective. Following prior work (Haarnoja et al., 2018; Eysenbach et al., 2022), we define the the entropy regularized goal-conditioned reward function as $\tilde{r}_g(s_t, a_t) \triangleq (1 - \gamma)\delta(s_t = g) - \alpha \log \pi(a \mid s, g)$, where $\delta(\cdot = g)$ is the delta measure at the goal $g$. Given a set of goals sampled from a goal distribution $g \sim p_{\mathcal{G}}(g)$, this new reward function allows us to rewrite the objective of the goal-conditioned policy as maximizing the entropy-regularized discounted state occupancy measure: $\max_\pi \mathcal{L}_{\text{Actor}}(\pi)$, where $\mathcal{L}_{\text{Actor}}(\pi)$ is expressed as:

$$\mathbb{E}_{\substack{g \sim p_{\mathcal{G}}(g) \\ \tau \sim \pi(\tau|g)}} \left[ (1 - \gamma) \sum_{t=0}^{\infty} \gamma^t \left( r_g(s, a) - \alpha \log \pi(a \mid s, g) \right) \right] \tag{1}$$

$$= \mathbb{E}_{\substack{g \sim p_{\mathcal{G}}(g), s \sim \rho(s) \\ a \sim \pi(a|s,g), \\ s_f \sim p_\gamma^\pi(s_f = g|s,a,g) \\ a_f \sim \pi(a_f|s_f, g)}} \left[ \delta(s_f = g) - \alpha \log \pi(a_f \mid s_f, g) \right] \tag{2}$$

$$\approx \mathbb{E}_{\substack{g \sim p_{\mathcal{G}}(g) \\ s \sim p^\beta(s) \\ a \sim \pi(a|s,g)}} \left[ \exp(f_{\phi,\psi}(s, a, g)) - \alpha \log \pi(a \mid s, g) \right] \tag{3}$$

$$= \tilde{Q}_g(s, a) \tag{4}$$

Thus, we augment CRL to optimize the soft Q function $\tilde{Q}_g(s, a)$ by optimizing the CRL loss $\min_{\phi,\psi} \mathcal{L}_{\text{Critic}}(\phi, \psi)$, where:

$$\mathcal{L}_{\text{Critic}}(\phi, \psi) = \mathbb{E}_{\mathcal{B}} \left[ -\sum_{i=1}^{|\mathcal{B}|} \log \left( \frac{e^{f_{\phi,\psi}(s_i, a_i, g_i)}}{\sum_{j=1}^{K} e^{f_{\phi,\psi}(s_i, a_i, g_j)}} \right) \right.$$
$$\left. - \sum_{i=1}^{|\mathcal{B}|} \log \left( \frac{e^{f_{\phi,\psi}(s_i, a_i, g_i)}}{\sum_{j=1}^{K} e^{f_{\phi,\psi}(s_j, a_j, g_i)}} \right) \right] \tag{5}$$

for $\left\{ f_{\phi,\psi}(s_i, a_i, g_j)_{i,j} \right\}$ over the elements of the batch $\mathcal{B}$ (Bortkiewicz et al., 2025). The critic function $f_{\phi,\psi}(s, a, g)$ estimates expected discounted future state occupancy, and the actor objective combines this with an additional term $\mathcal{L}_{\text{Entropy}}(\theta) = -\mathbb{E}_{g \sim p_{\mathcal{G}}(g), \tau \sim \pi(\tau|g)} \left[ (1 - \gamma) \sum_{t=0}^{\infty} \gamma^t (\alpha \log \pi(a \mid s, g)) \right]$ that estimates expected discounted future entropy. This term will be optimized with temporal difference updates. See Appendix B for more details on the algorithm.

## 4.2. Variational Goal Inference

Following the motivation of Section 3.3, we will learn a variational distribution $q_\xi(g|\tau)$ to match the true posterior $p^\star(g|\tau)$. We optimize the forward KL objective to achieve this (Ambrogioni et al., 2019; Yu et al., 2019):

$$\min_\xi D_{KL}\left(p^\star(g \mid \tau) \| q_\xi(g \mid \tau)\right) \tag{6}$$

$$= \min_\xi \mathbb{E}_{p^\star(g,\tau)} \left[ \log \frac{p^\star(g \mid \tau)}{q_\xi(g \mid \tau)} \right] \tag{7}$$

$$= \max_\xi \mathbb{E}_{g \sim p(g); \tau \sim p^\star(\tau|g)} \left[ \log q_\xi(g \mid \tau) \right] \tag{8}$$

$$= \max_\xi \mathcal{L}_{\text{Info}}(\xi) \tag{9}$$

When our policy is trained to optimality, it will emit a trajectory distribution equivalent to $p^\star(\tau|g)$. Thus, we can use our online learned MaxEnt RL policy to sample trajectories both for contrastive RL pre-training and for learning the variational posterior.

Another way to model the variational posterior is with the mean field approximation: $q_\xi(g|\tau) = \prod_{t=0}^{T} q_\xi(g|s_t, a_t)$, where each local state-action independently influences the distribution over the goal. This form can be much easier to train since parameters $\xi$ are now shared across state-action inputs. We can rewrite the expression for the true posterior as $p^\star(g \mid \tau) = \frac{p^\star(\tau|g)p(g)}{p(\tau)} \propto p(g)e^{\sum_{t=0}^{T} r_\theta(s_t, a_t, g) - \frac{1}{T} \log Z_\theta} \propto \prod_{t=0}^{T} e^{r_\theta(s_t, a_t, g) - \frac{1}{T} Z_\theta}$, and note that it precisely takes a mean field form when the input trajectory is finite. Thus, we can establish a corollary to motivate the use of the mean field approximation

when optimizing $\mathcal{L}_{\text{Info}}(\xi)$ for our method, training a Gaussian MLP to perform amortized variational inference with the mean field approximation.

**Corollary 4.1.** *Without loss of generality, the class of mean field goal inference models includes the true posterior distribution.*

### 4.3. CIRL is Consistent

Our main theoretical result is to show that our method infers the correct distribution over expert goals. This statement is non-trivial because the most-frequented states may not be the user's intended state, so correctly performing goal inference requires reasoning about the relative difficulty of different goals. Proof can be found in Appendix A.1.

**Lemma 4.2.** *Let policy $\pi_{demo}$ be given. CIRL produces policy $\pi_{CIRL}$ that consistently infers rewards by converting the MaxEnt IRL problem into a goal inference problem:* $\min_\theta \mathbb{E}_{p(g)}\left[D_{KL}(p_E(\tau|g) \| p^\star(\tau|g)\right] \implies \max_\xi \mathbb{E}_{g \sim p(g); \tau \sim p^\star(\tau|g)}\left[\log q_\xi(g \mid \tau)\right]$

**IRL with FB is Inconsistent** FB (Touati & Ollivier, 2021) is presented as a method that can learn optimal policies for any task and proposes to imitate trajectories by inferring their reward and then using the corresponding reward-maximizing policy. We show that even if FB learns optimal policies for every reward function, it doesn't correctly identify which reward function a demonstrator is maximizing, thereby provably failing to perform zero-shot imitation. Proof can be found in Appendix A.2.

**Lemma 4.3.** *There exists an MDP with two unique reward-maximizing policies ($\pi_1, \pi_2$), where FB incorrectly demonstrates policy $\pi_1$ with policy $\pi_2$.*

### 4.4. Goal-Sampling

During training, we use states stored in the replay buffer to continually fit a Gaussian Kernel Density estimator (KDE) approximating the distribution of visited states. This buffer is pre-filled at the start of training with data from a randomly initialized policy, and at each iteration, we select the state from the buffer that has the lowest probability under the KDE to train the policy. We call this method of automatic exploration: GoalKDE. See Appendix B for a summary of the full CIRL algorithm.

## 5. Experiments

Our method contains components for self-supervised RL pretraining, automatic goal sampling, and goal inference. We ablate each in turn, and show that CIRL (CRL Pretraining + GoalKDE Exploration + Mean Field Goal Inference Model) can learn good representations for imitation across several environments. We use the JaxGCRL and

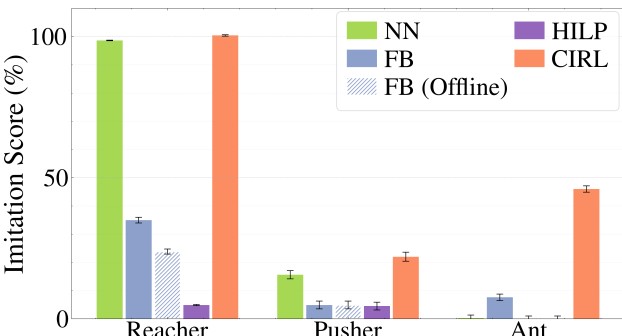

*Figure 2.* **Value of self-supervised RL pre-training** CIRL consistently outperforms the alternative FB representation zero-shot imitation method as well as the naive 1-NN policy baseline.

Unsupervised Reinforcement Learning Benchmark (URLB) environments (Bortkiewicz et al., 2025; Laskin et al., 2021). Details are provided in Appendix C.

For our evaluation, we train an expert policy using CRL under oracle goal sampling. Using this policy, we sample 2000 goals from the oracle test distribution of goals and unroll the CRL expert policy toward each goal. For each expert demonstration, we perform zero-shot IL across our ablation setup, reporting imitation score (the ratio between the cumulative return of the algorithm and the average cumulative reward of the expert) (Pirotta et al.). Unless otherwise noted, all methods were trained online. The only exceptions are FB (Offline) and goal-conditioned behavioral cloning (GCBC), which use the CRL expert policy data. FB (Offline) uses 2000 distinct trajectories of 1025 steps each sampled from the CRL expert policy. GCBC trains a goal-conditioned policy with the same number of update steps as the CRL expert. We also further evaluate using non-goal-conditioned policy demonstrations trained with URLB rewards on the Ant Forward, Ant Jump, and Ant Flip tasks and demonstrate the capability of CIRL to imitate these policies with low regret.

### 5.1. CIRL with Self-Supervised Pretraining Outperforms Baselines

We first compare CIRL against several baselines for imitation learning, including those with and without access to expert data during training. For each environment, we compared the reward earned by an expert policy (CRL) and the imitation learning method, reporting the fraction of expert reward achieved as the "imitation score." The baselines, both trained with no access to expert information, include the Nearest Neighbor baseline, which in a given state considers the 1-NN state in the expert demonstration and applies its corresponding action. We also include the URL baselines of FB representation (Touati & Ollivier, 2021) and HILP

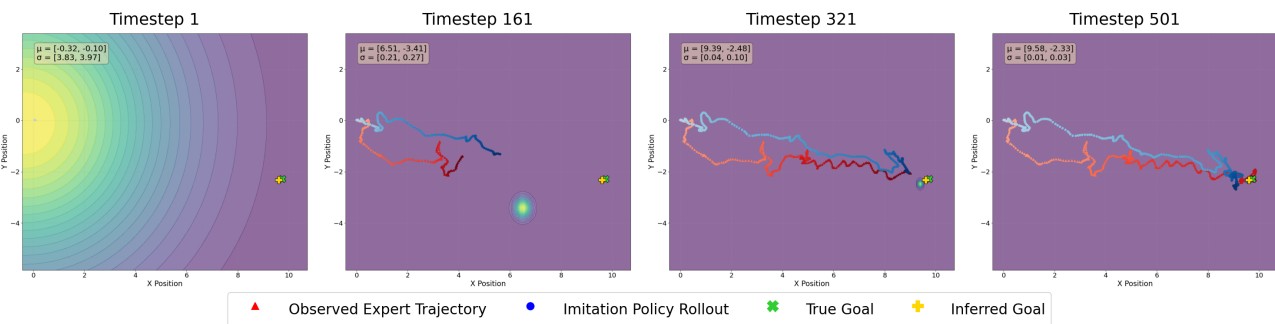

*Figure 3.* **Zero-shot imitation learning with CIRL via goal inference.** CIRL combines goal-conditioned contrastive RL pre-training, automatic goal sampling for exploration, and a mean field goal inference model to imitate expert demonstrations. Here we see how an Ant's imitation policy and posterior distribution over goal states evolve across timesteps toward a final maximum a posteriori (MAP) estimate.

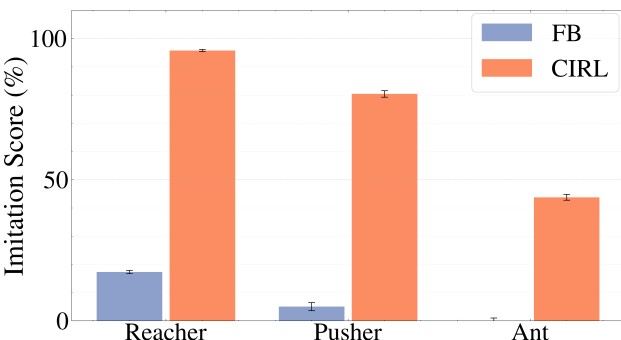

*Figure 4.* **Summarizing behavior via goals yields better imitation than reward-based explanations.** When using the last expert demonstration state as the goal, CIRL achieves high imitation scores on goal-conditioned environments while FB struggles to infer goal-conditioned reward functions from online learning.

(Wu et al., 2018). The inferred latents for these methods were computed from expert demonstration states (Pirotta et al.). As seen in Figure 2, CIRL consistently outperforms all baselines, regardless of environment difficulty, making it the most promising technique for learning to imitate in unfamiliar goal-conditioned environments. Even if we train the FB representation with data from the expert policy, this baseline is unable to achieve comparable imitation scores to our method.

### 5.2. CIRL Pre-Training Outperforms the FB Representation

CIRL and FB representation's algorithms have two main structural differences: the way it learns the successor representation and the way it infers intentions. To better understand why CIRL outperforms the FB representation, we hold the method of inferring intentions constant and only use information from the last state of the expert demonstration. Note that for tasks where the goal state is transient

(e.g. tossing a ball to reach a particular height), the last state in a trajectory may not contain enough information about the true goal, but for Ant, Reacher, and Pusher, the agents are able to reach and stay at all possible goals. As seen in Figure 4, FB only achieves a small fraction of the imitation score of CIRL under these conditions. This result provides evidence that, given the same training time budget, using CIRL to learn successor representations for reaching goals is easier than using FB to learn more successor representations for any general reward function. Additionally, in Figure 10 in Appendix D, even if we instead use the CIRL inferred goal to compute a latent for the FB representation, we are still not able to match CIRL's performance, though performance does improve compared to using the average backward representation of the demonstration or the backward representation of the last state.

### 5.3. Mean Field Approximation Improves Goal Inference

Our theory suggests that inferring goals using a mean field approximation should preserve predictive power compared to using the full $\tau$ as input to the context encoder. We also have fewer parameters to train under the mean field assumption, and thus hypothesize that it will outperform the full $\tau$ alternative. Testing this across environments with CIRL and GCBC, in Figure 5, we see that mean field goal inference universally outperforms the alternative of inferring goals, regardless of environment or training algorithm. These experiments validate our corollary of the preserved predictive power of mean field goal inference, with the added computational benefits of this simplified modeling choice. The mean field assumption also allows us to reliably infer goals from partial trajectories, as shown in Figure 3, where we see the posterior distribution hone in on the true goal as the imitator observes more of the demonstration.

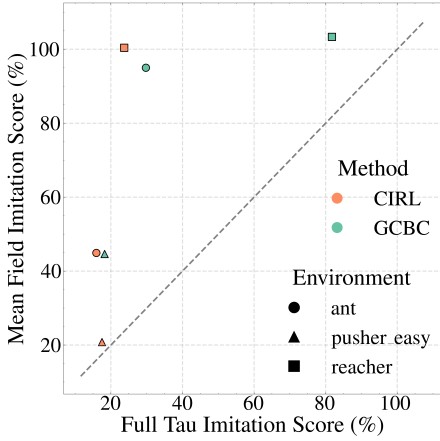

*Figure 5.* **Mean field goal inference models outperform alternative full $\tau$ input models.** Using a mean field form yields experimental performance gains without loss of generality.

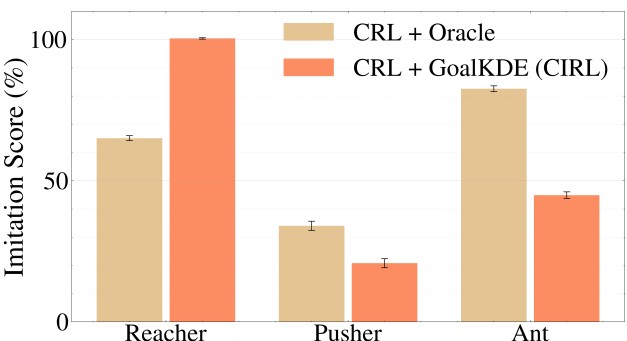

*Figure 6.* **GoalKDE exploration vs. oracle goal sampling during CRL pre-training.** Holding the goal inference method constant (mean field inference), we find that GoalKDE sampling can achieve a significant fraction of imitation score compared to the oracle baseline, and can even outperform this baseline in some environments.

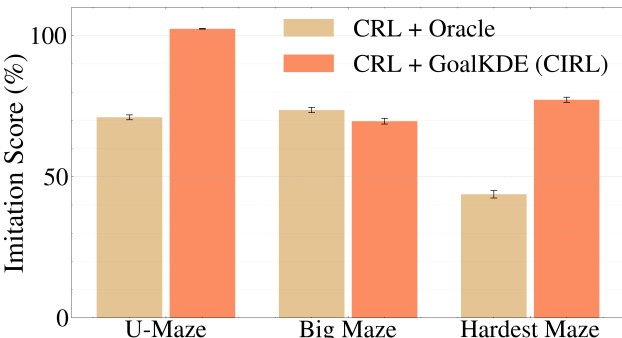

*Figure 7.* **Testing the limits of GoalKDE as a goal-sampling method for CIRL.** We see that the performance gap between CIRL and CRL + Oracle widens as the state space of PointMaze environments grows.

In Figure 7, we perform further analysis of GoalKDE's capabilities across increasingly complex PointMaze environments, with U-Maze having the smallest state space and Hardest Maze having the largest. We also increase the number of training timesteps accordingly as the Maze environment grows in complexity, as per Table 6. We see that as the state space grows, GoalKDE allows CRL to gracefully take advantage of increased training time to maintain performance, while CRL with oracle goal sampling fails. Our number of training timesteps is limited by training stability with the default hyperparameters. See Figure 9 in Appendix D for additional results ablating CIRL's goal inference method.

## 5.5. CIRL Supports Imitation Beyond Goal-Conditioned Environments

We run further experiments on the standard URLB benchmark, which is not designed for goal-reaching, to show that CIRL outperforms prior methods for zero-shot imitation when imitating policies (1) trained with more general reward functions and (2) which require expanding the goal hypothesis space. Following the URLB Benchmark, we train expert policies on the Ant Forward, Ant Jump, and Ant Flip tasks with PPO on non-goal-conditioned reward functions, and report regret of CIRL inferred policies compared to these expert policies. We see in Figure 8 that CRL pre-training methods can achieve lower regret than FB or HILP imitation policies by inferring goals involving the ant's torso 3D position and linear/angular velocity. CRL + Oracle goal sampling could perform better in some environments due to sampling fewer infeasible goals, and extensions to CIRL's exploration scheme based on related work could overcome this difficulty (OpenAI et al., 2021). Thus, CIRL can scale to more complex reward functions as long as we sufficiently expand the goal space to capture the task.

## 5.4. Better Automatic Goal Sampling Improves Imitation Scores

While we see that CIRL with GoalKDE automatic goal sampling outperforms our baselines with no expert data, we ablate our GoalKDE goal sampling method against oracle goal sampling (which trains CRL on the test-time goal distribution) to experimentally quantify the gap between these methods. We see in Figure 6 that in the Reacher environment, training CRL policies with GoalKDE can yield near-perfect imitation scores, and that sometimes GoalKDE can better explore the state space for more generalizable policies. However, for the higher dimensional state spaces in Ant and Pusher, a combination of more sophisticated goal sampling techniques or more training steps could boost performance beyond oracle sampling.

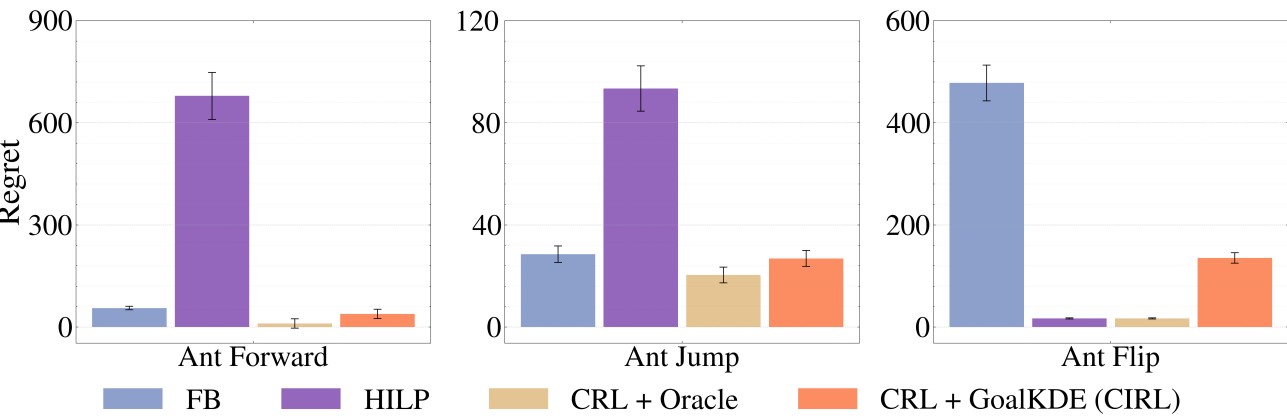

*Figure 8.* **CIRL inferred goals efficiently summarize complex rewards**. CIRL achieves lower regret than FB and HILP baselines when imitating URLB policies with non-goal-reaching rewards.

## 6. Limitations

Since not all reward functions are goal reaching, future work could close the gap between these reward hypothesis classes by exploring richer goal representations, such as language or multi-modal spaces, and consider summarizing behavior with multiple sub-goals. Our method also requires access to a simulator and would require further research to evaluate applicability in safety-critical settings, low data settings, or human interaction settings. With more complex goal spaces, related work in exploration could be applied as a substitute for our GoalKDE method. A full comparison of goal-sampling methods is outside of the scope of this paper. Our main aim is to propose a full pipeline for enabling imitation via an imagine-and-practice loop in the complete absence of expert data.

## 7. Conclusion

We introduced a framework for goal-conditioned maximum entropy inverse reinforcement learning that leverages self-supervised contrastive RL pretraining, automatic goal sampling, and a mean field variational goal inference model to enable zero-shot imitation from a single demonstration. By re-framing reward inference as goal state inference and coupling this with CRL, our method learns transferable policies across diverse task distributions. Our results support a favorable re-evaluation of the expressiveness of goals as a summary of behavior and crucially proves theoretically consistent imitation.

## Acknowledgements

This material is based upon work supported by the National Science Foundation under Award No. 2441665 and was partially funded by NSF OAC-2118201. Any opinions, findings and conclusions or recommendations expressed in this material are those of the author(s) and do not necessarily reflect the views of the National Science Foundation. The authors thank Prof. Ryan Adams for helpful discussions on the proofs in this paper. We thank Cathy Ji and Royina Jayanth for providing feedback on drafts of this paper. We also thank the members of Princeton RL Lab and LIPS Lab for feedback on iterations of this project. The authors are also pleased to acknowledge that the work reported in this paper was substantially performed using the Princeton Research Computing resources at Princeton University which is consortium of groups led by the Princeton Institute for Computational Science and Engineering (PIC-SciE) and Office of Information Technology's Research Computing.

## Impact Statement

This paper presents work whose goal is to advance the field of Machine Learning. There are many potential societal consequences of our work, none which we feel must be specifically highlighted here.

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

# A. Theoretical Analysis

## A.1. CIRL is Consistent

*Proof.* MaxEnt IRL corresponds to the following objective:

$$\arg\min_{\theta} D_{\mathrm{KL}}\left(p_{\pi_E(\tau)}\|p^\star(\tau)\right) = \arg\max_{\theta} \mathbb{E}_{p_{\pi_E}(\tau)}\left[\log p^\star(\tau)\right]$$

Under MaxEnt modeling, each goal g induces a trajectory model $p^\star(\tau \mid g) \propto \left[\rho_0(s_0)\prod_{t=0}^{T} p(s_{t+1} \mid s_t, a_t)\right]\exp\left(\sum_{t=0}^{T} r_g(s_t, a_t)\right)$ with log-partition $\log Z_g$. In a goal-conditioned setting, taking the reward to be entirely determined by g means the family $\{p^\star(\tau \mid g)\}_g$ is indexed by goals, and the learning objective can be posed as minimizing the average forward KL

$$\min_{\theta} \mathbb{E}_{p(g)}\left[D_{\mathrm{KL}}\left(p_E(\tau \mid g)\|p^\star(\tau \mid g)\right)\right],$$

where $p(g)$ is the goal prior used both in data collection and modeling.

Define the expert and model joints over $(\tau, g)$ as $p_E(\tau, g) = p(g)p_E(\tau \mid g)$ and $p^\star(\tau, g) = p(g)p^\star(\tau \mid g)$. When the same prior $p(g)$ is used, the average conditional KL equals a joint forward KL :

$$\mathbb{E}_{p(g)}\left[D_{\mathrm{KL}}\left(p_E(\tau \mid g)\|p^\star(\tau \mid g)\right)\right] = \mathbb{E}_{p(g)}\left[D_{\mathrm{KL}}\left(p_E(\tau, g)\|p^\star(\tau, g)\right)\right],$$

by applying Bayes Rule and canceling the identical priors.

Apply the KL chain rule to the joint KL:

$$D_{\mathrm{KL}}\left(p_E(\tau, g)\|p^\star(\tau, g)\right) = D_{\mathrm{KL}}\left(p_E(\tau)\|p^\star(\tau)\right) + \mathbb{E}_{\tau \sim p_E(\tau)}\left[D_{\mathrm{KL}}\left(p_E(g \mid \tau)\|p^\star(g \mid \tau)\right)\right],$$

Thus our MaxEnt IRL objective is

$$\min_{\theta} \mathbb{E}_{p(g)} D_{\mathrm{KL}}\left(p_E(\tau \mid g)\|p^\star(\tau \mid g)\right) = \min_{\theta}\left\{D_{\mathrm{KL}}\left(p_E(\tau)\|p^\star(\tau)\right) + \mathbb{E}_{p_E(\tau)} D_{\mathrm{KL}}\left(p_E(g \mid \tau)\|p^\star(g \mid \tau)\right)\right\} \quad (10)$$

Now we note that our marginal distribution $p^\star(\tau) = \int p(g)p^\star(\tau|g)dg$ is a difficult integral to compute and thus apply variational inference by introducing the amortized variational distribution $q_\xi(g; \tau)$. Then

$$\log p^\star(\tau) = ELBO(\theta, \xi; \tau) + D_{KL}(q_\xi(g|\tau)\|p^\star(g|\tau))$$

where

$$ELBO = \mathbb{E}_{q_\xi}\left[\log p(g) + \log p^\star(\tau|g) - \log q_\xi(g|\tau)\right]$$

Taking the expectation over expert trajectories:

$$\min_{\theta}\left\{D_{\mathrm{KL}}\left(p_E(\tau)\|p^\star(\tau)\right)\right\} = \max_{\theta} \mathbb{E}_{p_E(\tau)}\left[\log p^\star(\tau)\right]$$

$$= \max_{\xi} \mathbb{E}_{q_\xi}\left[\log p(g) + \log p^\star(\tau|g) - \log q_\xi(g|\tau)\right]$$

$$= \min_{\xi}\left[D_{KL}(q_\xi(g|\tau)\|p^\star(g|\tau))\right]$$

Now we see the major issue with using the ELBO/reverse KL is that it requires us to be able to evaluate the conditional likelihood $p^\star(\tau|g)$. This is impossible in our scenario, but we could sample from it since we can sample from the trajectory

distribution of our MaxEnt RL policy. This motivates the use of **Forward Amortized Variational Inference (FAVI)**, which uses the forward KL instead of the reverse KL in its optimization (Ambrogioni et al., 2019).

The loss function of FAVI derives from the joint-contrastive variational inference objective and is expressed as:

$$\mathcal{L}_{\text{FAVI}}[p, q] = D(p^{\star}(g, \tau) \| q_\xi(g, \tau))$$

To approximate the intractable posterior $p^{\star}(g \mid \tau)$, we factorize the variational joint as the product of a variational posterior $q_\xi(g \mid \tau)$ and a sampling distribution of the data:

$$q_\xi(\tau, g) = q_\xi(g \mid \tau)k(\tau)$$

Now we note:

$$D_{KL}(p^{\star}(\tau, g) \| q_\xi(\tau, g)) = \mathbb{E}_{p^{\star}(\tau, g)} \left[ \log \frac{p^{\star}(\tau, g)}{q_\xi(g \mid \tau)k(\tau)} \right] \tag{11}$$

$$= -\mathbb{E}_{p^{\star}(\tau, g)}[\log q_\xi(g \mid \tau)] + \mathbb{E}_{p^{\star}(\tau, g)} \left[ \log \frac{p^{\star}(\tau, g)}{k(\tau)} \right] \tag{12}$$

Considering only the terms that depends on $q$, we can define the FAVI loss as follows:

$$\mathcal{L}_{\text{FAVI}} = -\mathbb{E}_{p^{\star}(\tau, g)}[\log q_\xi(g \mid \tau)]$$

This is precisely the loss function $\mathcal{L}_{\text{Info}}(\xi)$ we train. Therefore, for our goal-conditioned setting, the IRL problem can be reduced to one of learning a variational posterior with FAVI. Importantly, note that the partition function is implicit within the samples we generate from the joint distribution via $g \sim p(g), \tau \sim p^{\star}(\tau \mid g)$, allowing us to consistently infer goals where methods that ignore the partition function do not. □

### A.2. FB is Inconsistent

We prove this by providing a counterexample. The key idea in the counterexample is that an infrequently visited state may nonetheless be the policy's desired goal. We illustrate this with a simple 2-state MDP.

*Proof.* We define an MDP with 2 states $(s_1, s_2)$ and 2 actions $(a_1, a_2)$ with the following dynamics:

$$p(s' \mid s, a) = \begin{cases} s_1, & \text{if } s = s_1, a = a_1 \\ s_1, & \text{w.p. } \frac{1}{2} \text{ if } s = s_1, a = a_2 \\ s_2, & \text{w.p. } \frac{1}{2} \text{ if } s = s_1, a = a_2 \\ s_2, & \text{if } s = s_2 \end{cases}. \tag{13}$$

Assume that the initial state is distributed $p_0(s) = \mathbb{1}(s_1)$. Note that state $s_2$ has just one action. The only decision to make is the action at initial state $s_1$. Since all MDPs have deterministic optimal policies, there are just two unique (potential) reward-maximizing policies for this MDP:

$$\pi_1(a \mid s) = \begin{cases} a_1 & \text{if } s = s_1 \\ \text{any action} & \text{if } s = s_2 \end{cases} \tag{14}$$

$$\pi_2(a \mid s) = \begin{cases} a_2 & \text{if } s = s_1 \\ \text{any action} & \text{if } s = s_2 \end{cases} \tag{15}$$

We will show that when data are collected from policy $\pi_2$, FB infers that data were collected with policy $\pi_1$. This policy is clearly different, achieving different amounts of rewards (for all non-trivial reward functions).

We next compute the occupancy measure for policy $\pi_2$. From the initial state $s_1$, the policy transitions to state $s_2$ with probability $\frac{1}{2}$ at each time step. Thus, the probability of still being at state $s_1$ after $t$ time steps decays as $1/2^t$. The occupancy measure can thus be written as:

$$\rho^{\pi_2}(s = X) = (1 - \gamma) + [1 + \gamma\frac{1}{2} + \gamma^2\frac{1}{2^2} + \gamma^3\frac{1}{2^3} + \cdots] \tag{16}$$

$$= (1 - \gamma)\sum_{t=0}^{\infty}(\gamma/2)^t = \frac{1 - \gamma}{1 - \gamma/2}. \tag{17}$$

Then $\rho^{\pi_2}(s = Y) = 1 - \rho^{\pi_2}(s = X)$. Thus, when $\gamma$ is small enough, policy $\pi_2$ "spends more time at" state $x$ than state $y$:

$$\gamma < \frac{2}{3} \implies \rho^{\pi_2}(s = s_1) > \rho^{\pi_2}(s = s_2). \tag{18}$$

This will be a problem for FB, which infers rewards based not on the difficulty of maximizing them, but rather instead based on visitation counts. FB infers rewards via:

$$z_R = \sum_t B(s_t). \tag{19}$$

Without loss of generality, we assume that $B(s_t) = \mathbb{1}(s_t)$, a one-hot vector; this solution is always admissible if the representations have high-enough dimension. Thus, the inferred reward function is

$$r(s) = \begin{cases} \frac{1-\gamma}{1-\gamma/2} & \text{if } s = s_1 \\ \frac{\gamma/2}{1-\gamma/2} & \text{if } s = s_2 \end{cases} \tag{20}$$

Note that state $s_1$ has a higher reward than state $s_2$ with $\gamma < \frac{2}{3}$. Thus, the reward-maximizing policy for this reward function is $\pi_1$ (which stays in $s_1$), not $\pi_2$ (which sometimes transitions to the lower reward state $s_2$). $\qquad\square$

This demonstrates that FB incorrectly identifies demonstrations from $\pi_2$ as coming from $\pi_1$. The fundamental issue is that FB uses the occupancy measure directly as the reward signal without considering the partition function or the policy's optimality under that reward. This leads to systematic misidentification of the demonstrator's true policy.

## B. Algorithm

We present pseudocode for training our zero-shot IL method based on contrastive RL pretraining:[1]

---

[1]Code released at: https://github.com/kwantlin/contrastive-irl.

**Algorithm 1** Contrastive IRL

1: **Input:** CRL loss $\mathcal{L}_{\text{Critic}}$ and energy function $f_{\phi,\psi}(s,a,g) = \phi(s,a)^T\psi(g)$ (Eysenbach et al., 2022), Entropy-regularization value function $\mathcal{L}_{\text{Entropy}}$, actor objective $\mathcal{L}_{\text{Actor}}$, variational posterior loss $\mathcal{L}_{\text{info}}$
2: Initialize $\phi, \psi, \theta, \xi, \pi$ and a pre-filled replay buffer $\mathcal{D}$
3: **repeat**
4:     **in parallel over environments**
5:         $g = \arg\min_g KDE(\mathcal{D})$
6:         Store $\tau \sim \pi(s,g)$ in $\mathcal{D}$

7:     **for** $j = 1, \ldots,$ `num_updates` **do**
8:         Randomly sample (with discount) a batch $\mathcal{B}$ from $\mathcal{D}$ of state-action pairs and goals from their
            future
9:         Update critic:
            $(\phi, \psi) \leftarrow (\phi, \psi) - \alpha\nabla_{\phi,\psi}\big[\mathcal{L}_{\text{Critic}}(\mathcal{B}; \phi, \psi)\big]$
10:       Update entropy-regularization value function:
            $(\theta) \leftarrow (\theta) - \alpha\nabla_\theta\big[\mathcal{L}_{\text{Entropy}}(\mathcal{B}; \theta)\big]$
11:       Update policy:
            $\pi \leftarrow \pi - \alpha\nabla_\pi\big[\mathcal{L}_{\text{Actor}}(\mathcal{B}; \phi, \psi, \pi)\big]$
12:       Update variational posterior:
            $q \leftarrow q - \alpha\nabla_\xi\big[\mathcal{L}_{\text{Info}}(\mathcal{B}; \phi, \psi, \pi)\big]$

13: **until** convergence

## C. Experimental Details

We ran our experiments building off the JaxGCRL benchmark (Bortkiewicz et al., 2025). Unless otherwise mentioned, we used the same hyperparameters as that implementation. $\alpha$ used for Maximum Entropy IRL was 1e-5. For the FB representation and HILP baselines, we use the same encoder networks as in JaxGCRL and the same actor and critic learning rates. The metric value net for the HILP baseline similarly uses the JaxGCRL encoder architecture for the phi network. For the context encoder, we also use the JaxGCRL encoder and train to predict the mean and variance of a Gaussian. The backbone MLP of the JaxGCRL encoder networks has a hidden width of 256 units, hidden depth of 2 layers, and 1 skip connection. We use the Swish activation after each hidden Dense layer. We used the seed 2 to train each type of policy.

*Table 1.* Reacher environment hyperparameters

| hyperparameter | value |
| --- | --- |
| batch size | 1024 |
| num timesteps | 20,000,000 |
| num environments | 256 |

*Table 2.* Pusher environment hyperparameters (goal: 3D position and 3D linear velocity)

| hyperparameter | value |
| --- | --- |
| batch size | 256 |
| num timesteps | 60,000,000 |
| num environments | 512 |

*Table 3.* Ant environment hyperparameters (goal: 2D position)

| hyperparameter | value |
|---|---|
| batch size | 512 |
| num timesteps | 30,000,000 |
| num environments | 1024 |

*Table 4.* Ant environment hyperparameters (goal: 3D position and 3D linear velocity)

| hyperparameter | value |
|---|---|
| batch size | 256 |
| num timesteps | 600,000,000 |
| num environments | 512 |
| healthy z range | (0.0, 4.0) |
| target z | Uniform over range (0.2, 2.0) |
| target 3D linear velocity | Uniform over (-1.0, 1.0) |

*Table 5.* Ant environment hyperparameters (goal: 3D angular velocity)

| hyperparameter | value |
|---|---|
| batch size | 256 |
| num timesteps | 600,000,000 |
| num environments | 512 |
| target 3D angular velocity | Uniform over (-2.5, 2.5) |

*Table 6.* PointMaze environment hyperparameters (goal: 2D position)

| hyperparameter | value |
|---|---|
| batch size | 1024 |
| num environments | 256 |
| num timesteps | 40,000,000 (U-Maze) |
|  | 100,000,000 (Big Maze) |
|  | 120,000,000 (Hardest Maze) |

## C.1. Environments

**Reacher:** This environment is a 2D manipulation task involving a two-jointed robotic arm. The goal is to move the arm's end effector to a sampled 2-dimensional target located randomly within a workspace disk. The 11-dimensional state space includes joint angles and velocities along with the position of the end effector. The 2-dimensional action represents torques applied at the arm's hinge joints.

**Pusher:** This features a 3D robotic arm and a movable object resting on a surface. The objective is to push the object into a 2D goal location randomly sampled at each episode reset. The 23-dimensional state space includes the arm's joint angles, velocities, and the position of the movable object. The 7-dimensional action space controls the robotic arm via continuous motor torques at its joints.

**Ant:** This locomotion task involves a quadruped navigating towards target XY positions randomly sampled from a circle around its starting position. The 29-dimensional state space comprises the robot's joint positions, orientations, and velocities, and the 8-dimensional action space consists of torques applied to each of the multiple leg joints. When using CIRL to infer URLB rewards, we expand the goal space to include the 3D position and 3D linear velocity or 3D angular velocity.

**PointMaze:** This navigation task involves a point mass navigating towards a target XY position in a constructed maze. The 4-dimensional state space includes the position and linear velocity of the point mass, and the 2D action space controls linear force along slide joints of the point mass in the environment. The possible configurations include U-Maze (20 x 20 maze layout), Big Maze (32 x 32 maze layout), and Hardest Maze (36 x 48 maze layout), in order of increasing state space size.

## D. Additional Results

Figure 9 ablates CIRL performance against alternate goal inference methods: knowing the true goal or inferring the goal to be the last state of the expert demonstration. When we provide the true goal to a policy trained with CRL and GoalKDE exploration, we get a significant boost in imitation score, with any gap in imitation score from 100% likely due to distribution shift between goals sampled via GoalKDE and those from the oracle test distribution, and alternative methods for goal exploration are a promising area for future work in GCRL. For goal-conditioned expert policies, inferring the last state to be the goal can be a strong baseline, but would fail when we try to imitate a task such as an Ant jumping.

In Figure 10, we imagine if the FB representation had access to the CIRL mean field inferred goal. Using this state to compute a backward representation latent to command the FB policy, FB is still not able to match the performance of CIRL. This is likely because FB must perform more exploration to learn a zero-shot policy for all possible rewards, including goal-reaching rewards, while CIRL has a smaller hypothesis space of rewards to explore and learn. However, we do note that FB with the inferred goal does outperform FB with the latent computed from either the last state of the demonstration or the averaged backward representation of all states in the demonstration.

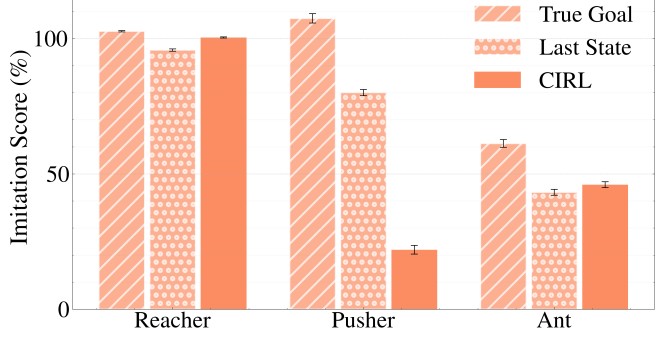

*Figure 9.* **Ablating CIRL.** For Pusher, most of the performance gap is due to goal inference, but for the Ant environment, most of the performance gap is likely due to distribution shift induced by GoalKDE.

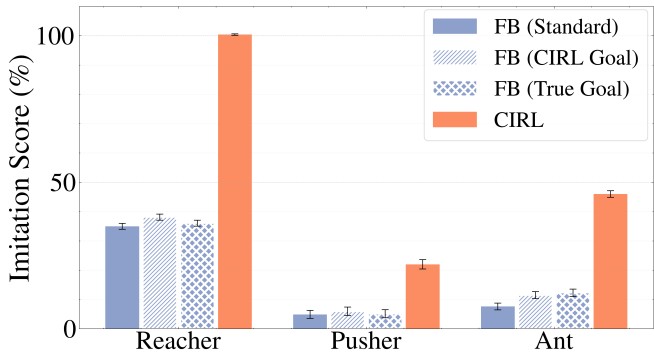

*Figure 10.* **Use the backward representation of the CIRL inferred goal to command the FB policy.** Comparing alternate methods of computing the FB latent against the standard FB baseline, CIRL still achieves higher imitation scores.

