# OpenReview forum: "Consistent Zero-Shot Imitation with Contrastive Goal Inference"
_ICML.cc/2026/Conference — ICML 2026 regular_

### Official Review · Reviewer_pBxi · 2026-02-25

**Soundness:** 3
**Presentation:** 3
**Significance:** 3
**Originality:** 3
**Overall Recommendation:** 4
**Confidence:** 3

**Summary:**

This paper proposes Contrastrive Inverse Reinforcement Learning (CIRL), and addresses zero-shot Imitaiton Learning (IL) for goal-conditioned tasks where the agent is given a single demonstration and do not conduct any gradient update or trial-and-error. It consists of three steps: 1) self-supervised contrastive RL pretraining to learn soft (maximum entropy-based) goal-conditioned policy and Q-value, 2) a goal inference model to learn the goal posterior through KL minimization, and 3) automatic goal sampling during pretraining by fitting a Gaussian kernel density estimator and selects the lowest probability state in a buffer filled with randomly initialized policy. The proposed method is proved theoretically to correctly infer the distribution over expert goals, while prior work does not. On several environments, the proposed method outperforms baselines such as HB and HILP.

**Compliance With Llm Reviewing Policy:**

Affirmed.

**Final Justification:**

The response addressed most of my concerns, and overall I think this is a good work and I am positive towards this paper. However, I still feel that the number of data required to train the agent (even as the authors point out, spread on 1024 different goals) is a bit too much for potential real-life downstream applications. While this is not necessarily a ground for rejection, it could potentially limit the impact of the paper.

**Key Questions For Authors:**

1. Could the authors report the computational resource (e.g. GPU type and hours spent) for the proposed method? Judging from Tab. 1 to 6, the proposed method seems to be very expensive.

2. The authors claim that "the work extends CRL to the MaxEnt RL setting" (line 108-109), but there seems to be no ablations on the effect of MaxEnt RL. Could the authors report the performance gain from using MaxEnt RL and/or the effect of the hyperparameter for soft Q-learning?

**Limitations:**

Yes.

**Strengths And Weaknesses:**

**Strengths**

1. The paper is well-written and easy to follow; the high-level idea that a goal-based policy is pretrained via a maximum-entropy contrastive RL with self-proposed goals is very clearly conveyed.

2. The authors provide code and detailed hyperparameters, which shows good reproducibility of this work. The experiment result also looks solid with visualizations on how it works in a 2D plane and ablations such as those in Fig. 8, Fig. 9 and Fig. 10. It is particularly interesting that FB with CIRL goal can outperform standard FB.

3. The proposed idea is intuitive, and also well-supported by theoretical justifications; I appreciate the in-depth theoretical analysis on how FB (the baseline) is inconsistent and unable ot identify which reward function a demonstrator is maximizing, while the proposed method is consistent.

**Weaknesses**

1. In line 47-48, the authors claim that "generative models are primarily built by optimizing self-supervised objectives on input data collected offline by humans". I would disagree with this statement - this could be true in 2022-2023, but LLM researchers has realized the importance of scaling up reinforcement learning for a long time, at least ever since the success of OpenAI o1 and DeepSeek-R1 (with DeepSeek-R1-zero that has 100% post-training to be reinforcement learning). Grok-4 is well-known to have pretraining-level of computation resources invested on reinforcement learning. It is not "unclear whether today's recipe for building generative AI foundation models will be directly applicable to interactive settings" (line 44-45) - the interactive recipe is widely accepted and heatedly focused on by the LLM community [1].

2. While the related work is very detailed, it does not introduce the relation of the proposed work to existing Contrastive RL (CRL) works, which I believe is one of the most important components of this work.

3. The pretraining of the proposed method seems quite costly; the model needs to be trained for tens of millions or even hundreds of millions of step (Tab. 1 to Tab. 6 in the appendix) to work on a proprioceptive environment, which is a huge number compared to the usual amount of steps for RL works trained on these environments (about 10^5-10^7). While I understand that zero-shot generalization to any goal is impressive, this amount of data could make actual application of the proposed method prohibitively expensive.

**Minor Weaknesses**

1. I would suggest to add a teaser figure in parallel to the zero-shot imitation learning to show the procedure of the proposed algorithm.

2. There are large blank area left in some parts of the paper due to formatting, e.g. the blank around Fig. 2.

3. The year for the reference to Pirotta et al. (Fast imitation via behavior foundation models) is missing.

**References**

[1] D. Silver and R. Sutton. Welcome to the Era of Experience. https://storage.googleapis.com/deepmind-media/Era-of-Experience%20/The%20Era%20of%20Experience%20Paper.pdf.

---

> ### Author Rebuttal · Authors · 2026-03-31
>
> We thank the reviewer for their thoughtful comments. We appreciate that they found our paper “well-written and easy to follow”, our proposed idea “well-supported by theoretical justifications”, and our experimental results “solid”. To address key comments:
>
> > Q1: statements on generative models/LLMs
>
> We thank the reviewer for this important point and fully agree that the framing could be updated. The success of o1, DeepSeek-R1, and the scaling of RL in post-training have made clear that interactive learning is already central to building frontier AI systems. We also appreciate the pointer to Silver and Sutton's "Era of Experience" position paper, which articulates this shift well. We will revise the intro in the camera-ready version.
>
> However, we believe the core motivation of our paper remains relevant. In the LLM setting, the agent still receives a human-designed reward signal (verifiable correctness, human preferences, or outcome supervision). In our setting, the agent has no reward signal whatsoever and must propose its own objectives. To our knowledge, this self-supervised explore-then-imitate paradigm has no direct analogue in the LLM RL pipeline.
>
> We will rewrite the introduction to reposition our contribution more precisely. The challenge we address is not whether interactive learning matters, but how an embodied agent can conduct self-supervised interactive pre-training that prepares it for zero-shot imitation of arbitrary tasks.
>
> > Q2: comparison to existing CRL works
>
> We will add the following citations and discussion of related work on the CRL algorithm for scaling properties [1], offline RL [2], and extensions to visual (pixel-based) scenes and combinatorial optimization problems [3].
>
> > Q3: computation cost
>
> Each policy trained in the paper used an Nvidia H100 GPU. The goal-conditioned Ant, Reacher, Pusher, and Maze environments trained for 4-12 hours and URLB experiments for 24 hours. Time needed for convergence in these environments may indeed be much lower but we set a conservative upper limit to ensure convergence. For example, note in the Ant environment, training for 30 million steps is counted cumulatively across 1024 parallel environments, each with a single goal/task. Therefore, we are roughly spending 30k steps per task, which is within the range of 10^5-10^7 steps per task per environment. The JaxGCRL benchmark we used allowed us to massively parallelize and speed up these steps and training iterations, meaning low cost for multi-task learning and inverse RL.
>
>
> > Q4: performance gain from MaxEnt RL
>
> Like previous works in inverse RL, we adopt the MaxEnt assumption to obtain an analytic expression for the probability of a trajectory given a goal, which allows us to derive a procedure for goal/reward inference. While on some RL tasks, learning MaxEnt policies helps, sometimes simply learning to maximize reward is most robust. Whether MaxEnt RL gives us performance gains on our imitation tasks is orthogonal to this work, but the MaxEnt assumption is necessary for inferring rewards tractably. The value of alpha may be tuned per task [4] in some works or tuned to meet a target entropy based on the action dimension [5], but we choose a fixed value across experiments.
>
> [1] Wang, Kevin, Ishaan Javali, Michał Bortkiewicz, Tomasz Trzcinski, and Benjamin Eysenbach. 2025. “1000 Layer Networks for Self-Supervised RL: Scaling Depth Can Enable New Goal-Reaching Capabilities.” In The Thirty-Ninth Annual Conference on Neural Information Processing Systems.
>
> [2] Zheng, Chongyi, Benjamin Eysenbach, Homer Walke, Patrick Yin, Kuan Fang, Ruslan Salakhutdinov, and Sergey Levine. 2023. “Stabilizing Contrastive RL: Techniques for Robotic Goal Reaching from Offline Data.” arXiv [Cs.LG]. arXiv. https://doi.org/10.48550/arXiv.2306.03346.
>
> [3] Park, Seohong, Kevin Frans, Benjamin Eysenbach, and Sergey Levine. 2024. “OGBench: Benchmarking Offline Goal-Conditioned RL.” arXiv [Cs.LG]. arXiv. https://doi.org/10.48550/arXiv.2410.20092.
>
> [4] Haarnoja, Tuomas, Aurick Zhou, Pieter Abbeel, and Sergey Levine. 2018. “Soft Actor-Critic: Off-Policy Maximum Entropy Deep Reinforcement Learning with a Stochastic Actor.” arXiv [Cs.LG]. arXiv. https://doi.org/10.48550/arXiv.1801.01290.
>
> [5] Haarnoja, Tuomas, Aurick Zhou, Kristian Hartikainen, George Tucker, Sehoon Ha, Jie Tan, Vikash Kumar, et al. 2018. “Soft Actor-Critic Algorithms and Applications.” arXiv [Cs.LG]. arXiv. https://doi.org/10.48550/arXiv.1812.05905.

---

> > ### Author Rebuttal · Reviewer_pBxi · 2026-04-01
> >
> > Thanks for the detailed response. The response addressed most of my concerns, and overall I think this is a good work and I am positive towards this paper. However, I still feel that the number of data required to train the agent (even as the authors point out, spread on 1024 different goals) is a bit too much for potential real-life downstream applications. While this is not necessarily a ground for rejection, it could potentially limit the impact of the paper.

---

### Official Review · Reviewer_eGt5 · 2026-03-06

**Soundness:** 2
**Presentation:** 1
**Significance:** 2
**Originality:** 3
**Overall Recommendation:** 4
**Confidence:** 2

**Summary:**

This work introduces a contrastive inverse reinforcement learning algorithm for training interactive agents for zero-shot imitation.

In the pretraining stage, a goal-conditioned RL policy is trained online in the environment. The goals are obtained in a self-supervised manner by sampling from a replay buffer using Gaussian kernel density estimation to identify less visited states.
During Inference, an expert trajectory is given to the model in order to estimate the expert's goal for the rollout.

The method is evaluated on the JaxGCRL benchmark in three environments: Reacher, Pusher and Ant.

**Compliance With Llm Reviewing Policy:**

Affirmed.

**Final Justification:**

The rebuttal helped to clarify the contribution of this work. The proposed idea seems good, but other reviewers raise valid concerns. Overall, I raised my score towards accepting this work at this stage.

**Key Questions For Authors:**

As mentioned above, I am struggling to grasp the core contribution of the proposed framework. I will be looking closely at the other reviews. But maybe you could try to clarify if my understanding of the method so far has been correct.

**Limitations:**

yes, but for clarity I recommend clearly separating the list of limitations from the conclusion in Sec. 6.

**Strengths And Weaknesses:**

I am struggling to identify and understand the core contribution.

In my view, reframing the inverse reinforcement learning problem of reward inference to a goal state inference problem seems like a strong contribution.

However, I believe the work would strongly benefit from improving its clarity:
- The abstract does not make it clear to me what the work is about. Most of the abstract until about l.026 is more confusing than helping to grasp the contributions.
- Similar with Fig. 1, which does not help to understand the problem setting, and is never referenced in the main text. I believe a "system overview" figure showing the three method components and their place in the training / inference pipeline would significantly help to understand the method.
- If I understand correctly, the self-supervised goal sampling approach for the RL pretraining is a novel core component of the proposed method. However, I am unsure if this component has been sufficiently investigated. The experiment in Sec. 5.4 seems very limited with only three environments, of which two seem to perform worse with the proposed GoalKDE method. The derived conclusion that better sampling leads to better policies also seems limited in its insight. I recommend increasing the number of evaluated tasks, and baseline methods, e.g. uniform random sampling of visited states.


Minor Comments:
- I recommend using numbered equation blocks for the multi-line equations in Sec. 3. Additionally, all (!) equations blocks should be numbered so that future works can easily refer to them.
- The introduction and related work sections are very extensive. However, I believe clarity could be improved by making these sections more concise. The comparisons and differentiation of the proposed method to existing approaches is especially helpful, but is currently getting lost within the text.

---

> ### Author Rebuttal · Authors · 2026-03-31
>
> We thank the reviewer for their thoughtful suggestions, and for the comment that “reframing the inverse reinforcement learning problem of reward inference to a goal state inference problem seems like a strong contribution”. We agree that this is our core contribution here, in addition to our proof of the method’s consistency. The proof of FB’s inconsistency we provide in parallel should motivate the necessity of ensuring other related and future works in zero-shot RL truly produce theoretically consistent imitation.
>
> To address comments and improve clarity of presentation, we address the following concerns:
>
> > Q1: Abstract clarity
>
> We appreciate the feedback on improving the abstract’s clarity, and suggest this new version:
>
> Zero-shot imitation learning requires an agent to reproduce expert behavior from a single demonstration without additional environment interaction or gradient updates at test time. We introduce Contrastive Inverse Reinforcement Learning (CIRL), a self-supervised framework that pre-trains interactive agents to instantly imitate novel demonstrations. CIRL rests on a key observation: when expert behavior can be summarized by a goal state, the multi-task inverse RL problem reduces to goal inference. During pre-training, CIRL jointly learns three components without any rewards, demonstrations, or preferences: (1) a maximum-entropy goal-conditioned policy via contrastive RL, (2) an automatic goal proposal mechanism (GoalKDE) that drives exploration, and (3) a mean-field variational model that performs amortized goal inference from trajectories. We prove that this procedure consistently recovers the demonstrator's intended goal by accounting for the relative difficulty of reaching different states and show how structurally similar prior work may otherwise fail to infer the correct behavior. At test time, CIRL infers the expert's goal from a single demonstration and commands the pre-trained policy with no further learning. Experiments on goal-conditioned and standard reward-maximizing control tasks show that CIRL outperforms prior zero-shot imitation methods, supporting the expressiveness of goals as a compact summary of behavior.
>
>
> > Q2: GoalKDE analysis
>
> We agree that the GoalKDE is important to extending CRL to true zero-shot imitation settings. To that end, we note that our results comparing imitation policies with Oracle Sampling against those with GoalKDE sampling consist not only of the 3 main goal-conditioned environments shown in Figure 6 (referenced in Section  5.4) but also the URLB environments in Figure 7. In addition, we reference an additional Figure 10 in the appendix with results on a set of Maze environments to stress test the performance of GoalKDE in increasingly large state spaces. For each Maze environment, Oracle and GoalKde policies saw the same number of env steps, but with GoalKDE exploration, policy performance is far more robust as the maze grows larger. For simpler tasks, GoalKDE may lag slightly due to lack of test-time task access, but for the more difficult URLB and larger maze tasks, it allows for far stronger exploration of large state spaces.
>
> > Q3: Formatting and figures
>
> We agree with the reviewers suggestions for improved spacing and equation numbering and Figure 1 improvements, and will include these in the final paper.

---

> > ### Author Rebuttal · Reviewer_eGt5 · 2026-04-01
> >
> > I sincerely thank the authors for their patient and detailed response.
> >
> > After reading the reviews and rebuttals I will update my review towards accepting this work.

---

### Official Review · Reviewer_TcyL · 2026-03-07

**Soundness:** 3
**Presentation:** 4
**Significance:** 3
**Originality:** 3
**Overall Recommendation:** 4
**Confidence:** 3

**Summary:**

The authors propose a novel zero-shot IL method, based on combining contrastive learning with IRL (CIRL). Training is self-supervised - the agent proposes goals, tries to reach them, and learns from this; no demos, rewards or preferences are needed. During testing, given a single demo, the learned IRL module infers the latent goal, then the learned goal-conditioned policy is used to reach it, zero-shot. The authors show that their method outperforms related methods on the URLB environments.

**Compliance With Llm Reviewing Policy:**

Affirmed.

**Final Justification:**

Given the rebuttal, I will be keeping my score the same.

**Key Questions For Authors:**

- I can understand the task setting of having "no additional environment interaction" or "test-time data", but why not "gradient updates"? This might be impractical for fine-tuning foundation models, but should still be feasible with the smaller models used in this paper.
- The theory suggests that the mean field approximation of goals (Section 4.2), but this doesn't mean it should be better than performing inference over the full trajectories, as shown in Fig. 5? Can you provide some comments on this?
- The limitations list "access to a simulator", but I believe this is just for safety/sample efficiency reasons, and not about a requirement to reset the environment to particular states?

**Limitations:**

The authors do provide several relevant limitations.

**Strengths And Weaknesses:**

The introduction clearly motivates the research and outlines the contributions. The related work seems very comprehensive; I expect few people would be experts in all the areas covered, so I commend the authors for this section. The methodology does lay out the authors' framework, highlighting the general pipeline is, vs. specific implementation choices. The experiments show that the proposed method outperforms several baselines on a selection of environments, and contains careful studies of assumptions made in the methodology. Section 5.5 is important, as it shows that the proposed method extends beyond goal-conditioned environments, which is one of the main claims of the paper. I also appreciate Section A.2, with its analysis of FB's inconsistency/the importance of the partition function.

The main limitation of this work is the potential scaleability. Only 4 environments are tested, with the most complex having a 29D state space and 8D action space; these are all relatively simple continuous control environments. Empirically, does this work with pixel inputs? Can it extend to the sorts of environments hinted at in the motivation, e.g., Franka Kitchen?

---

> ### Author Rebuttal · Authors · 2026-03-31
>
> We thank the reviewer for their thoughtful comments and questions, especially their highlighting of key sections of this paper that “contain careful studies of assumptions made in the methodology.” We hope the clarity of presentation and analysis here motivates the contribution well. To address key comments:
>
> > Q1: Scalability
>
> While our paper focuses on providing initial empirical evidence and theoretical proof for a new framework of zero-shot imitation, we agree that scalability is key for extending this work. To that end, this requires scalability of CRL and of the goal inference mechanism. CRL scalability and stability have been recently explored with great promise, including within the JaxGCRL benchmark, to higher dimensional pixel-based observations, and in both online and offline RL settings [3,4,5]. For the goal-inference model, extensions of our model to diffusion or flow-based architectures could be employed. Works such as Zero Shot Visual Imitation and ENVISION have shown that inferring full visual goals from demonstrations is well within the capacity of today’s models. For automatic goal generation in high dimensional space, we note in limitations that more sophisticated goal-sampling strategies could be employed. For example, methods for unsupervised environment design (e.g. PAIRED) could be used to procedurally generate candidate goal images for training. For more complex tasks, applying our method on top of a pretrained behavioral foundation model would also greatly reduce training time necessary to acquire skills.
>
>
> > Q2: Why no gradient updates at test-time
>
> Our goal is to study how much an agent can prepare for imitation purely through self-supervised exploration without ever seeing expert data during training. Allowing gradient updates at test time would conflate two questions: (1) how good are the pretrained representations and policies, and (2) how effective is the fine-tuning procedure. The zero-shot setting isolates question (1), which is our contribution.
>
> Zero-shot inference is strictly harder, so any method that succeeds zero-shot will only improve with fine-tuning. By demonstrating strong zero-shot performance, we establish a lower bound on what self-supervised pretraining alone can achieve.
> Practical motivation does exist beyond model size and benchmarking. Even with small models, gradient-free inference is desirable when the agent must respond instantly to a new demonstration (e.g. real-time human-robot interaction) or only a single demonstration is available, making gradient updates prone to overfitting.
>
>
> > Q3: Mean field approximation
>
> Our ability to apply independence across time steps to our goal inference model is strictly derived from the probability of a trajectory given a goal in Section 3.2. This independence arises from the Markov assumption, but does not mean that we do not reason over the entire trajectory. In Figure 3, we see how the goal inference prediction changes with partial trajectories, and predicting the true goal does require reasoning over the entire trajectory. The independence assumption instead allows us a simplification that Figure 5 shows can have empirical benefits.
>
>
> > Q4: Simulator access
>
> Yes, simulator access is purely for safety and sample efficiency. Resets to particular states are never necessary.
>
> References:
>
> [1] Ziebart et al. 2008.
>
> [2] InfoGAIL
>
> [3] Wang et al. 2025 “1000 Layer Networks for Self-Supervised RL: Scaling Depth Can Enable New Goal-Reaching Capabilities.”
>
> [4] Zheng et al 2023. “Stabilizing Contrastive RL”
>
> [5] Park et al 2024 OGBench

---

> > ### Author Rebuttal · Reviewer_TcyL · 2026-04-02
> >
> > Thank you for the clarifications. Given the amount of components in the system, I believe that the experiments are insufficient to demonstrate that the proposed method can scale. Considering other reviews as well, I will be keeping my recommendation.

---

### Official Review · Reviewer_rxQP · 2026-03-09

**Soundness:** 3
**Presentation:** 2
**Significance:** 2
**Originality:** 2
**Overall Recommendation:** 3
**Confidence:** 3

**Summary:**

This paper propose a pipeline for enabling imitation via an imagine-and-practice loop in the complete absence of expert data, including goal-conditioned maximum entropy inverse reinforcement learning that leverages self-supervised contrastive RL pretraining,automatic goal sampling, and a mean field variational goal inference model to enable zero-shot imitation from a single demonstration.

**Compliance With Llm Reviewing Policy:**

Affirmed.

**Final Justification:**

Given the rebuttal, I keep my score unchanged.

**Key Questions For Authors:**

See in weaknesses

**Limitations:**

yes

**Strengths And Weaknesses:**

Strengths:
1)Self-supervised contrastive RL provides a convenient way to learn structured state–goal representations that integrate well with the proposed goal inference objective.
2)The formulation of imitation learning as goal posterior inference provides an elegant probabilistic perspective and theoretically guarantees that CIRL is consistent.
3)The mean-field goal inference formulation is  a computationally efficient approximation and preserves the predictive power compared to the full trajectory as input to the encoder.
4)Effective evaluation on standard benchmarks
Weaknesses:
1)The contrastive RL component  appears largely based on existing methods and does not introduce substantial algorithmic novelty.
2) The independence assumption across time steps may limit its applicability in tasks where goal inference requires reasoning over the entire trajectory.
3)While the method performs well on standard goal-reaching tasks, the zero-shot imitation learning claim implicitly assumes that the goal is identifiable from the trajectory. In settings where different intention can induce similar trajectories, the posterior p(g∣τ) may be multi-modal, and the method may fail to recover the intended goal,especially in one trajectory demonstration.
4)The evaluation tasks are relatively simple goal-reaching problems where universal goal-conditioned policies can be learned effectively with standard goal randomization.It is unclear whether the proposed goal inference mechanism provides substantial advantages in more complex imitation settings and hard to see that the proposed pipeline could  scale to more general cases.

---

> ### Author Rebuttal · Authors · 2026-03-31
>
> We thank the reviewer for their thoughtful comments. We appreciate that the reviewer notes the strengths of our work to “theoretically guarantee that CIRL is consistent” and to provide “effective evaluation on standard benchmarks.” We hope our results expose the experimental advantages of our approach over related methods. We also hope our proof of FB’s inconsistency can motivate the necessity of ensuring other related and future works in zero-shot RL truly produce theoretically consistent imitation.
>
> > Q1: Novelty
>
> While our choice of contrastive RL (CRL) as the goal-conditioned policy-learning algorithm is based on prior work, our work provides several key contributions on top.
>
> Firstly, we noted that while many inverse RL approaches today struggle with difficult reward identification problems, we demonstrate the advantages of approaching zero-shot imitation by inferring goal-conditioned rewards instead. This is especially clear in our experiments with the URLB benchmark in Figure 7. This evidence motivates a favorable re-evaluation of the expressiveness of goals as a summary of behavior.
>
> Secondly, given the strengths of goal-conditioned RL algorithms such as CRL, we also noticed a gap in attempts to use zero-shot RL methods for imitation. While related works such as FB, HILP, and others typically include some comments on how their methods can be applied to imitating expert trajectories, there has been insufficient investigation of the consistency of these proposed imitation schemes. In proving FB’s method inconsistent, we noted an important gap to fill in providing a consistent algorithm with our goal-conditioned approach.
>
> In addition the combination of contrastive RL with automatic goal sampling has not been explored; previous works assumed access to the test-time goal distribution (oracle goal sampling). Our pipeline introduces GoalKDE during self-supervised pre-training to effectively match or even exceed oracle goal sampling performance. Figure 10 in the appendix supports this with evidence on increasingly complex maze environments.
>
> > Q2: “The independence assumption across time steps may limit its applicability in tasks where goal inference requires reasoning over the entire trajectory.”
>
> Our ability to apply independence across time steps to our goal inference model is strictly derived from the MaxEnt IRL assumption, defining the probability of a trajectory given a goal in Section 3.2. This independence arises from the Markov assumption, but does not mean that we do not reason over the entire trajectory. In Figure 3, we see how the goal inference prediction changes with partial trajectories, and predicting the true goal does require reasoning over the entire trajectory. The independence assumption instead allows us a simplification that Figure 5 shows can have empirical benefits.
>
> > Q3: Identifiability of the goal
>
> We agree that identifiability of rewards from demonstrations is still a fundamental problem in inverse RL and imitation learning (i.e. many reward functions could be optimal for a given demonstration) [1]. Indeed, in our setting, this would translate to a multi-modal posterior p(g|tau). We note that this is a weakness inherent to the inverse RL problem rather than our paper specifically and extensions of our method to address the issue, including collecting additional demonstrations or mixture posteriors, could be easily incorporated [2].
>
> > Q4: Scalability
>
> While our paper focuses on providing initial empirical evidence and theoretical proof for a new framework of zero-shot imitation, we agree that scalability is key for extending this work. To that end, this requires scalability of CRL and of the goal inference mechanism. CRL scalability and stability have been recently explored with great promise, including within the JaxGCRL benchmark, to higher dimensional pixel-based observations, and in both online and offline RL settings [3,4,5]. For the goal-inference model, extensions of our model to diffusion or flow-based architectures could be employed. Works such as Zero Shot Visual Imitation and ENVISION have shown that inferring full visual goals from demonstrations is well within the capacity of today’s models. For automatic goal generation in high dimensional space, we note in limitations that more sophisticated goal-sampling strategies could be employed. For example, methods for unsupervised environment design (e.g. PAIRED) could be used to procedurally generate candidate goal images for training. For more complex tasks, applying our method on top of a pretrained behavioral foundation model would also greatly reduce training time necessary to acquire skills.
>
> References:
>
> [1] Ziebart et al. 2008.
>
> [2] InfoGAIL
>
> [3] Wang et al. 2025 “1000 Layer Networks for Self-Supervised RL: Scaling Depth Can Enable New Goal-Reaching Capabilities.”
>
> [4] Zheng et al 2023. “Stabilizing Contrastive RL”
>
> [5] Park et al 2024 OGBench

---

> > ### Author Rebuttal · Reviewer_rxQP · 2026-04-03
> >
> > Thanks for the clarification. The responses addressed most of my concerns. I still believe that the evaluation is insufficient to validate the scalability of the proposed method. I will keep my score unchanged.

---

### Decision · Program_Chairs · 2026-04-30

**Decision:**

Accept (regular)

**Comment:**

Overall, I recommend weak accept. The reviewers generally agree that the paper has a clear and elegant formulation of zero-shot imitation as goal inference, a meaningful theoretical contribution through the consistency result, and solid empirical performance on standard benchmarks. The main concerns are that the core RL machinery is largely built on existing contrastive RL ideas, and that the evaluation is still limited to relatively simple continuous-control settings, leaving scalability and broader applicability uncertain. Some of the reviewers updated their scores after the author response, while others had concerns that were not fully addressed.

Strengths:
+ Clear, intuitive framing of imitation as goal posterior inference.
+ Strong theoretical analysis, including the consistency argument and the critique of FB.
+ Solid experimental results and generally good presentation/reproducibility.

Weaknesses:
-  Limited novelty in the underlying contrastive RL component.
- Evaluation is narrow, mostly on simple goal-reaching environments, so scalability is unclear.
- The goal identifiability and mean-field assumptions may break in more complex or multimodal imitation settings.